# BOA/DHB/Na: An Efficient UV-MALDI Matrix for High-Sensitivity and Auto-Tagging Glycomics

**DOI:** 10.3390/ijms232012510

**Published:** 2022-10-19

**Authors:** Erina Barada, Hiroshi Hinou

**Affiliations:** 1Graduate School of Life Science, Hokkaido University, Sapporo 001-0021, Japan; 2Frontier Research Center for Advanced Material and Life Science, Faculty of Advanced Life Science, Hokkaido University, Sapporo 001-0021, Japan

**Keywords:** glycomics, MALDI, glycan, oligosaccharide, solid ionic matrix

## Abstract

Matrix selection is a critical factor for success in glycomics studies using matrix-assisted laser desorption/ionization–mass spectrometry (MALDI–MS). In this study, we evaluated and optimized a new solid ionic matrix—*O*-benzylhydroxylamine (BOA)/2,5-dihydroxybenzoic acid (DHB)/Na—containing BOA and a small amount of sodium as the counter salt of DHB. The concentration of a mixture of BOA/DHB/Na and glycans on a MALDI target plate led to *O*-benzyloxy tagging of the reducing ends of the glycans. The BOA/DHB/Na matrix showed excellent aggregation performance and the ability to form a homogeneous solid salt on the MALDI target plate with a water-repellent surface. In addition, the BOA/DHB/Na matrix showed a simple peak pattern with suppressed in-source and post-source decay of the reducing ends of the glycans, as well as improved ionization efficiency of glycans. Utilizing the characteristics of the BOA/DHB/Na matrix, *O*-glycan analysis of porcine stomach mucin showed excellent detection sensitivity and reproducibility of the peak patterns. This BOA/DHB/Na matrix can accelerate glycomics studies using MALDI–MS and, in combination with other organic salt-type matrices that we have developed, constitutes a valuable tool for glycomics studies.

## 1. Introduction

Protein glycosylation is a dynamic process that is dependent on the local and global environment of the cell and organelles and is critical for physiological and pathological cellular functions [1]. Dysregulation of the structural dynamism of glycans can be a potential biomarker of inflammation, infections, immune diseases, and cancer. Hyper-sialylation and hyper-branching glycans of tumor cell surfaces are established hallmarks of several cancers [2]. Therefore, the development of a rapid and highly accurate method for glycan analysis is essential for the elucidation and reliable diagnosis of diseases.

Matrix-assisted laser desorption/ionization–time-of-flight mass spectrometry (MALDI–TOFMS) is a simple, rapid, highly sensitive, high-resolution, and robust method for analyzing biomolecules, including glycans [3]. 2,5-Dihydroxybenzoic acid (DHB) [4] is a versatile matrix with salt tolerance and ionization capacity for a wide range of substrates and has become the gold standard for MALDI–TOF analysis of glycans. However, its ionization efficiency and selectivity for glycan are insufficient, especially for sialylated glycans, causing noise in the spectrum and making glycan profiling difficult [5]. To avoid this problem, the mass spectrometry of sialylated glycans requires the chemical modification of carboxylic acids and subsequent glycan purification processes [6,7].

A variety of liquid ionic matrices [8] and some solid ionic matrices [9,10] have been developed to improve the sensitivity and selectivity of glycan analysis. Recently, we developed solid ionic matrices from aniline/DHB/Na and *N*-methylaniline (NMA)/DHB/Na [10]. These matrices can efficiently ionize sialylated glycans without chemical modification of the carboxylic acid of the sialic acid residue. Unlike DHBs, which form heterogeneous large crystals, the DHB salts with aniline derivatives form an amorphous-like homogeneous solid surface suitable for automated analysis along with the analyte. Furthermore, these DHB salt matrices suppress peptide ionization and cause selective in-source decay at the reducing end of the glycans. These properties of the solid ionic matrices enable new rapid analytical techniques such as pseudo-MS/MS/MS glycan sequencing [10] and top-down glycan pattern analysis of intact glycoproteins [11]. Selective tagging of the reducing ends of glycans is also a fundamental technology for glycan analysis, and aniline selectively forms an imine at the reducing end of glycans, making it a potential tool for the discrimination of reducing oligosaccharides from glycoconjugates [9,10]. However, the low reproducibility of imine formation [12] makes the aniline/DHB/Na matrix unsuitable for large-scale glycomics.

In this paper, we report a new solid ionic matrix system, introducing *O*-benzylhydroxylamine (BOA) instead of aniline. Oximes are much more stable than imine due to the negative inductive effect of the high electronegativity of oxygen (i.e., α-effect) [13,14]. Hence, BOA is expected to be able to stably derivatize glycans [15]. In addition to their different reactivity, the structural similarity of BOA and aniline can provide an improved solid salt matrix. Accordingly, this study investigated the stabilization of product ions and the possibility of glycomics of unmodified sialylated and neutral oligosaccharides and *O*-glycans using reflectron mode MALDI–TOFMS and MALDI–TOF/TOFMS profiling with this new matrix system.

## 2. Results and Discussion

### 2.1. Evaluation of the BOA Matrix

A hydrochloride salt of BOA (BOA HCl), easily available as a stable solid, is commonly used as a tagging agent for glycomics studies [15,16,17,18,19,20]. However, pure BOA—not the hydrochloride salt—would be suitable for the use of BOA as a counter salt for DHB sensitivity. Therefore, matrices of both forms of BOA mixed with DHB and sodium bicarbonate at a 12:10:1 ratio—BOA HCl/DHB/Na and BOA/DHB/Na—were evaluated (Figure 1). In addition, two aniline salt matrices—aniline/DHB/Na and NMA/DHB/Na [10]—were also prepared by mixing aniline or *N*-methylaniline at the same ratio in place of BOA. We first tested the BOA matrices using a decasaccharide fragment (SGP-10) prepared from egg yolk sialylglycopeptide via endoglycosidase digestion as the analyte (Figure 1 and Appendix A).

First, the morphology of these four DHB salt-type matrices with 5 pmol of SGP-10 on the MALDI target plate was evaluated in comparison with the commonly used 0.1% trifluoroacetic acid (TFA) solution of DHB (Figure 2 and Appendix A). The DHB (0.1% TFA) matrix tended to form heterogeneous needle-shaped crystals, as previously reported in [4,16], while the pure BOA and BOA HCl matrices, as well as the aniline/DHB/Na and NMA/DHB/Na matrices, formed amorphous solids at the center spot of the AnchorChip^TM^ MALDI target plate via an ionic-liquid-like intermediate state [10]. This intermediate state is advantageous for forming small aggregation spots on water-repellent surfaces such as an AnchorChip^TM^.

As matrices for MALDI–TOFMS analysis of glycans, the BOA salts of DHB showed clear advantages over the acid form of DHB in terms of sensitivity and reproducibility, as did the aniline salt (Figure 3, Appendix A). Compared with the conventional acid form matrix, the DHB in 0.1% TFA, BOA/DHB/Na, and BOA HCl/DHB/Na matrices clearly gave a parent ion signal of SGP-10 as a sodium adduct with a superior signal-to-noise (S/N) ratio (BOA Pure; S/N = 260, BOA HCl; S/N = 77, DHB TFA; S/N = 26). Compared with BOA HCl/DHB/Na, the BOA/DHB/Na matrix suppressed the lack of sialic acid, and a clear increase in the reaction efficiency of *O*-benzyl oxime (BOA) tagging at the reducing end of SGP-10 was observed. As previously reported, the aniline/DHB/Na matrix showed a lower rate of imine formation at the reducing end of SGP-10, while the NMA/DHB/Na matrix showed no modification of the reducing end [10]. The BOA HCl/DHB/Na matrix and the aniline/DHB/Na matrix showed inhibitory effects on the lack of sialic acid following the modification of the reducing end of SGP-10. The BOA/DHB/Na and NMA/DHB/Na matrices were fully controllable on and off the reducing end modification of SGP-10, respectively, giving a simple sodium addition ion signal pattern.

The pH levels of the BOA/DHB/Na and BOA HCl/DHB/Na matrix solutions were 4.1 and 2.7, respectively. The pH of the BOA/DHB/Na matrix was close to the p*K*_a_ of protonated BOA (ca. 4.7), and the pH of the BOA HCl/DHB/Na matrix was close to the p*K*_a_ of DHB (2.97) [17]. This indicates that BOA forms salts with DHB and exhibits a buffering ability that stabilizes the pH of the solution to make it suitable for oxime formation. For these reasons, it is thought that the BOA/DHB/Na matrix was able to analyze unmodified sialyloligosaccharides with higher sensitivity, suppress the loss of SA, and stabilize modification of the reducing end.

### 2.2. Optimization of the BOA-Tagging and Solidification Time by the BOA/DHB/Na Matrix with Analytes

Although the ionic liquid intermediate of the BOA–DHB salt provided a reaction field for BOA-tagging of sugar chains and was advantageous for forming a homogeneous amorphous solid, the solidification time tended to be prolonged by the film of this intermediate during the blow-drying process of the mixture with an analyte [10]. Therefore, we optimized the oxime formation and solidification time with analytes using the BOA/DHB/Na matrix. First, the SGP-10 and BOA/DHB/Na matrix mixture was allowed to stand at room temperature. MALDI–TOFMS was performed after 1 h, 3 h, and overnight, showing that overnight was optimal (Figure 4a). To shorten the processing time, target plates with mixed droplets of SGP-10 and the BOA/DHB/Na matrix were placed in a thermostatic chamber for 1 h at 40 °C, 50 °C, and 60 °C, followed by MALDI–TOFMS, indicating that the 50 °C and 60 °C treatments were optimal. When the reaction time was reduced to 30 min, oxime formation was incomplete at 50 °C (Figure 4b, Appendix A). Mixed droplets of SGP-10 and the BOA/DHB/Na matrix formed an amorphous solid after standing at 50 °C for 30 min. This result indicates that the oxime formation reaction continued to proceed after solidification. When the mixture was allowed to stand at 60 °C, no clear difference was observed between 30 min and 1 h after solidification, confirming that hydrolysis of the sialic acid residues did not proceed. Comparison of the two conditions—room temperature overnight and 60 °C for 1 h—showed that the hydrolysis of sialic acid progressed more in the room temperature overnight condition. The evaporation of water due to heating promoted the oxime formation (i.e., dehydration) reaction and inhibited the hydrolysis of sialic acid. These results indicate that an incubation time of 30 min at 60 °C is required for the use of the BOA/DHB/Na matrix. For subsequent experiments, an incubation time of 1 h with a safety margin of 30 min was used.

### 2.3. Stabilizing Effect of the Reducing End of Oligosaccharides by BOA Modification from In-Source and Post-Source Decay in MALDI

Next, we used TOF/TOF analysis to investigate why the pure BOA/DHB/Na matrix suppressed the decay of the analyte ion and gave a simple parent ion pattern in unmodified glycan analysis in comparison with the MNA/DHB/Na matrix, in which no reducing end modification occurred (Figure 3a,d and Appendix A). Higher laser power gives higher internal energy to the analytes, leading to backbone fragmentation known as in-source decay (ISD) and/or post-source decay (PSD) [18,19]. Therefore, we set the laser irradiation power to 90% and measured the PSD pattern of the parent ion (*m*/*z* 2192) of BOA-tagged SGP-10 observed when using the BOA/DHB/Na matrix (Figure 5a and Appendix A). As a result, two Y-type fragment peaks (*m*/*z* 1879 and 1565) indicating sialic acid cleavage at the non-reducing end were observed as the main fragment ion, and no major peak indicating further fragmentation was observed. As previously reported in [10], TOF/TOF analysis of the parent ion (*m*/*z* 2087) of SGP-10 using an NMA/DHB/Na matrix under identical conditions revealed more pronounced A-type cross-ring cleavage at the reducing end of the GlcNAc residue in addition to non-reducing end sialic acid cleavage (Figure 5b and Appendix A). This difference in glycan fragmentation associated with the reducing-end modification was also observed in TOF/TOF analysis using BOA HCl/DHB/Na and aniline/DHB/Na matrices (Appendix A), and this fragmentation pattern was reflected in the MALDI–TOFMS spectrum under low laser irradiation energy (Figure 3 and Figure 4).

In other words, the BOA/DHB/Na matrix improved the ionization efficiency of glycans in MALDI–TOFMS observation and suppressed the fragmentation of molecular ions caused by the tagging function at the reducing ends of the glycans, giving simple glycan peaks with an excellent S/N ratio.

### 2.4. Profiling of O-Glycans from Mucin Using the BOA/DHB/Na Matrix

This nature of the BOA/DHB/Na matrix is suitable for profiling studies on mixtures of free glycans using MALDI–TOFMS analysis. Therefore, we carried out a proof-of-concept study using a simple scheme for *O*-glycan analysis via base excision [21,22] and Cotton hydrophilic interaction chromatography (HILIC) [23] separation (Figure 1). Mucin from porcine stomach (MPS) was treated with DBU and 50% hydroxylamine solution followed by the addition of an internal standard (IS; *N*,*N*′,*N*″,*N*‴-tetraacetylchitotetraose); the released glycan was captured on cotton wool and then eluted with water. Then, 0.5 µL of the Cotton HILIC eluate, theoretically equivalent to glycans from 100 ng of dried MPS, was mixed with each matrix, and MALDI–TOFMS measurements were performed (Figure 6a). Using the BOA/DHB/Na matrix, we identified 20 different *O*-linked glycans with BOA tags at the reducing end (Appendix A). Using the DHB/Na and DHB (0.1% TFA) matrices, only the standard internal glycan peaks were reproducibly detected from the same amount of the sample. Next, the *O*-glycan profiles in five lots of the same workflow in Figure 1 showed high reproducibility, with 10 major peaks (Figure 6b, Appendix A). The *O*-glycan profile of MPS detected by MALDI–TOFMS analysis using the BOA/DHB/Na matrix was consistent with the results of MALDI analysis using a multistep chemical treatment and separation process [24,25,26].

## 3. Materials and Methods

### 3.1. Materials and Reagents

Sialylglycopeptide (SGP) from egg yolk, the disialyldecasaccharide form of SGP (SGP-10), and *N*,*N*′,*N*″,*N*‴-tetraacetylchitotetraose were purchased from Tokyo Chemical Industry Co., Ltd. (Tokyo, Japan). Mucin from porcine stomach (MPS) type III was purchased from Sigma-Aldrich Corp. (St. Louis, MO, USA). Acetonitrile, aniline, *N*-methylaniline (NMA), 50% hydroxylamine solution, 2,5-dihydroxybenzoic acid (DHB), trifluoroacetic acid (TFA), 1,8-diazabicyclo [5.4.0]undec-7-ene (DBU), *O*-benzylhydroxylamine hydrochloride (BOA HCl), *O*-benzylhydroxylamine (BOA), and sodium bicarbonate were purchased from Wako Pure Chemical Industries, Ltd. (Osaka, Japan).

### 3.2. Matrices

First, 0.5 M DHB in CH₃CN/H_2_O (90:10, *v*/*v*) was diluted 10 times with TFA/CH₃CN/H_2_O at a ratio of 0.1:50:50 (*v*/*v*/*v*); 0.5 M DHB (10 μL) in CH₃CN/H_2_O (90:10, *v*/*v*) and 0.1 M sodium bicarbonate (5 μL) in water were adjusted to 100 μL with CH₃CN/H_2_O (50:50, *v*/*v*); 0.5 M DHB (10 μL) in CH₃CN/H_2_O (90:10, *v*/*v*), 1.0 M aniline (6 μL) in CH₃CN, and 0.1 M sodium bicarbonate (5 μL) in water were adjusted to 100 μL with CH₃CN/H_2_O (50:50, *v*/*v*); 0.5 M DHB (10 μL) in CH₃CN/H_2_O (90:10, *v*/*v*), 1.0 M NMA (6 μL) in CH₃CN, and 0.1 M sodium bicarbonate (5 μL) in water were adjusted to 100 μL with CH₃CN/H_2_O (50:50, *v*/*v*); 0.5 M DHB (10 μL) in CH₃CN/H_2_O (90:10, *v*/*v*), 1.0 M *O*-benzylhydroxylamine hydrochloride (6 μL) in CH₃CN, and 0.1 M sodium bicarbonate (5 μL) in water were adjusted to 100 μL with CH₃CN/H_2_O (50:50, *v*/*v*); 0.5 M DHB (10 μL) in CH₃CN/H_2_O (90:10, *v*/*v*), 1.0 M BOA (6 μL) in CH₃CN, and 0.1 M sodium bicarbonate (5 μL) in water were adjusted to 100 μL with CH₃CN/H_2_O (50:50, *v*/*v*). The pH of the matrix solutions was measured using an S2K922 pocket pH meter from ISFETCOM Co., Ltd (Saitama, Japan).

### 3.3. Preparation of Pre-Spotted MALDI Target Plates and Analytes

Matrix solutions (0.25 μL) were spotted on MTP AnchorChip^TM^ 400/384 TF plates (Bruker, Bremen, Germany), and the analyte solution (0.5 μL) was deposited on the pre-spotted plates. The morphology of the spotted matrices on the target plates was recorded using the built-in camera of the Ultraflex III MALDI–TOF/TOF instrument.

### 3.4. MALDI–TOFMS

An Ultraflex III MALDI–TOF/TOF instrument (Bruker; Bremen, Germany) equipped with a 200 Hz smartbeam Nd:YAG UV laser (355 nm) was used in positive reflectron mode. The spot position on the target plate was calibrated using an auto-teach function. Mass spectra for all samples were acquired in the range *m*/*z* 300–5000 for positive reflectron mode, with 500 laser shots within a random walk of the shooting position pattern at a laser frequency of 100 Hz. Human serum glycan [15] was used as an external standard for calibration. In LIFT-TOF/TOF mode, Bruker Peptide Calibration Standard II was used as an external standard for calibration. The generated ions were accelerated to a kinetic energy of 25.0 kV. In LIFT-TOF/TOF mode, the generated ions were accelerated to 8 kV and selected in a timed ion gate. The precursor ion and its fragment ions were further accelerated to 19 kV in the LIFT cell. In LIFT-TOF/TOF mode, mass spectra for all samples were acquired with 1200 laser shots at a laser frequency of 200 Hz. Metastable ions generated by laser irradiation were analyzed without any additional fragmentation process, e.g., collision energy. Unless otherwise specified above, the sample position and laser power employed the default settings prepared using FlexControl 3.5 software (Bruker; Bremen, Germany). The obtained mass spectra were further analyzed using the FlexAnalysis v. 3 software (Bruker; Bremen, Germany).

### 3.5. Activation of Cotton HILIC Microtips

To purify the *O*-linked glycans, Cotton HILIC microtips were activated as previously reported in [23]. A small piece of cotton wool (100% absorbent cotton) with a weight of approximately 1 mg was taken from a cotton wool pad and pushed into the end part of a 200 μL pipette tip using a blunt needle to create a Cotton HILIC microtip. Then, 200 μL of solvent A (90% acetonitrile, 0.1% TFA) was applied to a Cotton HILIC microtip, and 200 μL of water was applied followed by 200 μL of solvent A applied two more times.

### 3.6. O-Linked Glycan Analysis

*O*-linked glycans from MPS were released [22] and purified [23] with small modifications to the process. Briefly, 50% hydroxylamine solution (6 μL) and 15 μL of DBU were added to 10 μL of MPS solution (10 mg/mL). The mixture was heated at 37 °C for 75 min, followed by the addition of 1000 pmol of *N*,*N*′,*N*″,*N*‴-tetraacetylchitotetraose as an internal standard (I.S.), and the volume was adjusted to 200 μL with cold acetonitrile. The solution was applied to a Cotton HILIC microtip and eluted by pushing a micropipette. The eluted solution was collected and eluted again in the Cotton HILIC microtip. The Cotton HILIC microtip was washed twice with 200 μL of solution A and then twice with 50 μL of water to elute *O*-linked glycans from the Cotton HILIC microtip. The eluted solution was analyzed by MALDI–TOF without concentration. The detected *O*-linked glycans were quantified with a peak area of *N*,*N*′,*N*″,*N*‴-tetraacetylchitotetraose (I.S.) using FlexAnalysis v. 3 Software (Bruker, Bremen, Germany) and Microsoft Excel. *O*-linked glycan structural compositions were assigned by GlycoMod (ExPASy proteomics server, Swiss Institute of Bioinformatics: https://web.expasy.org/glycomod/, accessed on 17 September 2021) [20] using experimental masses and previous studies. 

## 4. Conclusions

To accelerate glycomics research, we optimized and demonstrated the performance of the BOA/DHB/Na matrix to simultaneously achieve on-target tagging of the reducing ends of glycans, efficient and selective generation of sodium adduct ions, and suppression of pseudo-ion generation by ISD and PSD of sialic acid residues and the reducing ends of glycans. The BOA/DHB/Na matrix, as well as the aniline/DHB/Na and NMA/DHB/Na matrices, improved the ionization efficiency of glycans while suppressing pseudo-ion formation by ISD and PSD on their reducing ends. Taking advantage of these changes in matrix properties associated with the choice of DHB counter salts, the aniline/DHB/Na matrix enables direct glycan analysis of glycoproteins, as we reported in [11], and the BOA/DHB/Na matrix can greatly improve glycomics research through the cleavage of glycans from glycoproteins and glycolipids, in addition to direct analysis of free glycan patterns. As a proof-of-concept test of this BOA/DHB/Na matrix, rapid and sensitive analysis of mucin *O*-glycans was achieved by simple manipulation via beta-elimination and HILIC separation.

## Data Availability

All data generated or analyzed during this study are included in this published article and its Appendix A.

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
