# Peer review of "BOA/DHB/Na: An Efficient UV-MALDI Matrix for High-Sensitivity and Auto-Tagging Glycomics"

_ijms, 2022, doi:10.3390/ijms232012510_

Round 1

Reviewer 1 Report

The manuscript submitted by Hinou reported the development of a new MALDI matrix composed of O-benzylhydroxylamine(BOA), 2,5-dihydroxybenzoic acid (DHB) and sodium bicarbonate (sodium source) for glycomics studies. The detailed condition optimizations are described and a clear evaluation comparison between the new matrix and the previous one developed by the same group is presented to show the significant improvement in mass spectrometry characterization of the selected analytes (SGP-10). The new matrix is also used for profiling of O-glycans from mucin and the results look impressive. There are only a few minor points that need to be addressed:

1. First paragraph of section 2.1 and figure 1. The description of the matrices is confusing. Four different combinations should be clearly described. Also the six chemicals in figure 1 are not equaled, DHB and sodium bicarbonate are in all four combinations, while the other four chemicals are in different combinations. 

2. Why 60 degrees +1 hour is selected as the best condition rather than 60 degrees +30 mins?

3. Line 157 and Line 163. Fig. S6 should be Fig. S6b.

In the supporting information:

1. Figure S5, the legend for the second figure from the bottom should be 30 min, 50 degrees.

2. Caption for figure S6a is missing.

3. Figure S7b should be replaced by high resolution graph.

Author Response

I appreciate your careful review and helpful comment to improve our manuscript. We hope this matrix is a potential choice for glycomics study for various researchers.

The manuscript submitted by Hinou reported the development of a new MALDI matrix composed of O-benzylhydroxylamine(BOA), 2,5-dihydroxybenzoic acid (DHB) and sodium bicarbonate (sodium source) for glycomics studies. The detailed condition optimizations are described and a clear evaluation comparison between the new matrix and the previous one developed by the same group is presented to show the significant improvement in mass spectrometry characterization of the selected analytes (SGP-10). The new matrix is also used for profiling of O-glycans from mucin and the results look impressive. There are only a few minor points that need to be addressed:

  1. First paragraph of section 2.1 and figure 1. The description of the matrices is confusing. Four different combinations should be clearly described. Also the six chemicals in figure 1 are not equaled, DHB and sodium bicarbonate are in all four combinations, while the other four chemicals are in different combinations. 

Yes. As you mentioned this Figure 1 is not appropriate to facilitate understanding of the combination of chemicals used in this study. I changed Figure 1 as following:

  1. Why 60 degrees +1 hour is selected as the best condition rather than 60 degrees +30 mins?

Yes, you are right. In our study, we adopted “60 degrees + 1 hour” but as you mentioned “60 degrees +30 mins” is the best condition. Shorter is better.

I changed the description at the end of section 2.2 as follows; “These results indicate that an incubation time of 30 minutes at 60 °C is required for the use of the BOA/DHB/Na matrix. For subsequent experiments, an incubation time of 1 hour with a safety margin of 30 minutes was used.”

  1. Line 157 and Line 163. Fig. S6 should be Fig. S6b.

Collected.

In the supporting information:

  1. Figure S5, the legend for the second figure from the bottom should be 30 min, 50 degrees.

The legend for the second figure in Figure S5 was corrected to 30 min, 50 degrees.

The legend for Figure S9a was also corrected to 1 hour, 60 degrees.

  1. Caption for figure S6a is missing.

I added the caption “S7a”.

  1. Figure S7b should be replaced by high resolution graph.

All MALDI-MS spectrum data in supplementary materials were replaced into high-resolution images.

English of the revised manuscript was checked by the English pre-editing service provided by MDPI, and I changed it following the edited result.

Reviewer 2 Report

Review on

BOA/DHB/Na (BOA Pure): an Efficient UV-MALDI Matrix for High-Sensitive and Auto-Tagging Glycomics

The authors developed and tested a novel ionic matrix, the mixture of 2,5-dihydroxybenzoic acid O-benzylhydroxylamine containing little sodium ion. The advantage of the new matrix was proven by the analysis of SGP-10 and porcine stomach mucin. The glycans were successfully derivatized by the BOA, allowing improved detection.

 The research is designed correctly, but experiments are missing to take such conclusions. The derivatization agents were published.

In summary, the article is not ready to publish in its present form.

In general, the title and the article suggest matrix development, however, the BOA acts as a derivatization agent. From the article, it is not clear whether the derivatization or the presence of the BOA molecule in the matrix is the reason for the improved detection.

It is misleading, which must be detailed in the article and of course, further measurements are required. (The derivatization of glycans with that molecule and the advantages were published before by Ying Zhang, Yuyang Zhu. DOI: 10.1177/2472630319898150)

The matrix was prepared by the mixing of the compounds, however, the efficiency of such a system can be evaluated if the matrix is synthesized and purified, obtaining the pure compound for the test. In other cases, the advantage of the application is speculative, may the presence of all compounds result in improved detection.

The derivatization should be carried out separately from the MALDI sample preparation to prove the relevance of the matrix. (It must be noted, that the in-situ derivatization has advantages if it is fast enough).

In MALDI sample preparation the selection and the amount of the applied matrix are crucial. In the case of such a mixture, the development of the sample preparation would be too complex, it would not worth working with it.

 The authors called their matrix BOA Pure, the name is misleading and means nothing. I suggest calling it a name explaining the composition.

I have some problems with the evaluation of the data. MALDI-TOF MS spectra are given, however, due to the non- or semi-quantitative nature of the MALDI source spectra alone means nothing. Many measurements are required and detailed statistical analysis must be done to take any conclusion on the advantage of such a matrix system.

Thus to make the results publishable a validation process must be carried out.

The long incubation time can be preferable for the derivatization, as suggested by the change of intensity ratios in figure 4. However, it must be mentioned that the long incubation time for the DHB matrix is not preferable. In figure 3 the lack of underivatized peaks in the a) b) and c) spectra suggests a long incubation time. If the same was used for the other matrices the spectra are not comparable since different time is optimal for them.

The most valuable part of the article is the “Profiling of O-glycans from mucin using BOA Pure matrix

“ part. Here the clear advantage of the used sample preparation is shown. In the introduction, it is written that the DHB is used commonly. However here the use of DHB resulted in no peaks. What is the reason? In the literature, it is written that it is not the best but possible to use, in the article based on the spectra it is useless. It must be cleared.

The comparison would be beneficial with other matrices with higher efficiency, which is better to use in glycan analysis.

Finally, rapid language editing may also be considered, although I am not a native speaker myself.

Author Response

The authors developed and tested a novel ionic matrix, the mixture of 2,5-dihydroxybenzoic acid O-benzylhydroxylamine containing little sodium ion. The advantage of the new matrix was proven by the analysis of SGP-10 and porcine stomach mucin. The glycans were successfully derivatized by the BOA, allowing improved detection.

 The research is designed correctly, but experiments are missing to take such conclusions. The derivatization agents were published.

In summary, the article is not ready to publish in its present form.

In general, the title and the article suggest matrix development, however, the BOA acts as a derivatization agent. From the article, it is not clear whether the derivatization or the presence of the BOA molecule in the matrix is the reason for the improved detection.

- I appreciate your critical comments and effort to provide them. In our study, BOA has two functions, improve the ionization efficiency and derivatization of glycan reducing end. As shown in Figures 4, 5, and 7, the ionization efficiency of analytes (SGP-10 and mucin-type glycans) was clearly improved in the experiment using BOA/DHB/Na compared to the commonly used DHB alone. In addition, comparing Fig. 4e with Fig. 5a, it can be seen that the spectrum of the un-BOA-labeled glycan portion of the same amount of SGP-10 appears more strongly than either spectrum measured with DHB. As shown in figure 4e and reported in reference 10 (Int. J. Mass Spectrom. 2019, 443, 109–115), 5 pmol/spot of SGP-10 is the detection limit for DHB matrix even in the presence of 0.1% TFA in the solution or 10 mol% sodium ion to DHB.

It is misleading, which must be detailed in the article and of course, further measurements are required. (The derivatization of glycans with that molecule and the advantages were published before by Ying Zhang, Yuyang Zhu. DOI: 10.1177/2472630319898150)

- Thank you for the publication information related to this study. I know this report as one of a previous study using BOA HCl. However, this report by Ying Zhang et al. does not report any of the earlier reports on BOA labeling and oxime labeling of glycans, nor does it mention any changes in ionization efficiency associated with BOA labeling. In this manuscript, we cited a typical example of BOA labeling of glycan for quantitative analysis systematically reported before the work by Ying Zhang et al. In conclusion, the work by Ying Zhang et al. should not refer in our study. Their publication is not pioneering and not referred actual pioneering works. As they described in their manuscript, the work by Ying Zhang et al. has a value for following the separation of the preparative amount of glycans, and additional modification (per-methylation) for conventional glycan analysis.

The matrix was prepared by the mixing of the compounds, however, the efficiency of such a system can be evaluated if the matrix is synthesized and purified, obtaining the pure compound for the test. In other cases, the advantage of the application is speculative, may the presence of all compounds result in improved detection.

- As described in the experimental section, all reagents used in this study, up to the standard glycans and specimens analyzed, were purchasable from reagent companies. We also describe the reagent concentrations, ratios, procedures, and processing conditions used in this study. We describe the exact amount of all reagents and analytes. In contrast to our report, Ying Zhang et al. did not report the exact amount of analyte in their MS study.

The derivatization should be carried out separately from the MALDI sample preparation to prove the relevance of the matrix. (It must be noted, that the in-situ derivatization has advantages if it is fast enough).

- As previously mentioned, comparing Fig. 4e with Fig. 5a, it can be seen that the spectrum of the non-BOA-labeled glycan portion of the same amount of SGP-10 appears more strongly than either spectrum measured with DHB. As shown in figure 4e and I reported in reference 10 (Int. J. Mass Spectrom. 2019, 443, 109–115), 5 pmol/spot of SGP-10 is the detection limit for DHB matrix even in the presence of 0.1% TFA in the solution or 10 mol% sodium ion to DHB. In our paper, we examine and optimize the reaction time of in-situ derivatization and the conditions to complete this modification reaction at a practical rate.

In MALDI sample preparation the selection and the amount of the applied matrix are crucial. In the case of such a mixture, the development of the sample preparation would be too complex, it would not worth working with it.

- We do not agree with the opinion that the method presented in this paper is "not worth" at all. As shown in section 2.4 "Profiling of O-glycans from mucin using BOA/DHB/NaBOA Pure matrix", we developed the BOA/DHB/Na matrix to simplify the sample preparation steps. If you mean that "the sample preparation" is matrix preparation, we disagree with this too. There is no analytical method that does not require the correct mixing steps of the reagents; MALDI-TOFMS usually requires the preparation of four reagents consisting of a matrix compound, TFA, and two solvents, mixed in the correct proportions.

 The authors called their matrix BOA Pure, the name is misleading and means nothing. I suggest calling it a name explaining the composition.

- I disagree with "means nothing" but agree that it can be misleading. I replaced the name "BOA Pure" with the matrix composition "BOA/DHB/Na" used at the beginning of the title.

I have some problems with the evaluation of the data. MALDI-TOF MS spectra are given, however, due to the non- or semi-quantitative nature of the MALDI source spectra alone means nothing. Many measurements are required and detailed statistical analysis must be done to take any conclusion on the advantage of such a matrix system.

Thus to make the results publishable a validation process must be carried out.

- In this study, we are performing a "semi-quantitative" analysis of porcine gastric mucin and testing its reproducibility by adding an internal standard. This testing using internal standards and reproducibility as indicators is a standard protocol in MALDI-TOFMS studies. We have already published several results of quantitative analysis of glycans by this methodology. Quantitative glycomics studies using BOA-labeled glycans have already been repeatedly reported, as represented by literature15, and this experiment is based on these findings.

The long incubation time can be preferable for the derivatization, as suggested by the change of intensity ratios in figure 4. However, it must be mentioned that the long incubation time for the DHB matrix is not preferable.

- As you pointed out, long incubation times are undesirable, so we have demonstrated that the labeling time can be reduced to 30 minutes as shown in figure 4. We have also demonstrated that hydrolysis of sialic acid can be suppressed in accordance with the shortened labeling time in aqueous solution. (Ying Zhang et al. used a labeling time of 6 hours in aqueous solution, and they did not investigate any effect on hydrolysis of sialic acid or other compounds as well as the sensitivity of the BOA-labeled compound for MS analysis.)

In figure 3 the lack of underivatized peaks in the a) b) and c) spectra suggests a long incubation time. If the same was used for the other matrices the spectra are not comparable since different time is optimal for them.

 - As mentioned in the text, the spectra in figures 3b and 3c clearly contain the "underivatized peaks" observed in figures 3d and e. I think only the spectrum of figure 3a can be called "the lack of underivatized peaks". Please reconfirm.

The most valuable part of the article is the “Profiling of O-glycans from mucin using BOA Pure matrix” part. Here the clear advantage of the used sample preparation is shown. In the introduction, it is written that the DHB is used commonly. However here the use of DHB resulted in no peaks. What is the reason? In the literature, it is written that it is not the best but possible to use, in the article based on the spectra it is useless. It must be cleared.

- Thank you for agreeing with the value of this part. In order to demonstrate the advantages of both the elimination of the labeling step and the increased sensitivity associated with the use of the BOA/DHB/Na matrix, the "Profiling of O-glycans" used the eluate after HILIC treatment without concentration. It is also clearly shown that the O-glycan solution used is derived from the original 100 ng of mucin. In this state, the concentration would be difficult to detect with the normal DHB matrix, so as the other cited papers indicate, a concentration operation is necessary.

To clarify this point, the following was added to Page 7, line 201: "from the same amount of the sample".

The comparison would be beneficial with other matrices with higher efficiency, which is better to use in glycan analysis.

- Comparative data with other matrices are shown in Figure 3. In addition, the mixture of aniline derivatives and DHB used for that comparison in Figure 3 was compared to a mixture of 3-AQ and CHCA, which we have now confirmed to have the maximum sensitivity for glycan detection, in a previous paper (Int. J. Mass Spectrom. 2019, 443, 109-115) has already been performed.

Finally, rapid language editing may also be considered, although I am not a native speaker myself.

- Thank you, I used the English pre-edit service provided by MDPI for my revised manuscript.

Round 2

Reviewer 2 Report

I accept the answers and suggest the corrected article for publication without further changes.

extra comments:

Rev: I have some problems with the evaluation of the data. MALDI-TOF MS spectra are given, however, due to the non- or semi-quantitative nature of the MALDI source spectra alone means nothing. Many measurements are required and detailed statistical analysis must be done to take any conclusion on the advantage of such a matrix system.
Thus to make the results publishable a validation process must be carried out.
Auth: In this study, we are performing a "semi-quantitative" analysis of porcine gastric mucin and testing its reproducibility by adding an internal standard. This testing using internal standards and reproducibility as indicators is a standard protocol in MALDI-TOFMS studies. We have already published several results of quantitative analysis of glycans by this methodology. Quantitative glycomics studies using BOA-labeled glycans have already been repeatedly reported, as represented by literature15, and this experiment is based on these findings.
comment of the reviewer:
The comparison of two MALDI-MS spectra is usually not enough to make a conclusion about the performance of the applied method. Simple statistical analysis (t-test, ANOVA, and so on…) would give clear insight. The intensity deviation of the MALDI spectra can be high, even with internal standards. Experienced MALDI-MS users can make difference. However for most readers, it is not trivial, or the selection of the proper spectra can be misleading.
In further studies, I suggest the use of statistical methods.

Rev: In figure 3 the lack of underivatized peaks in the a) b) and c) spectra suggests a long incubation time. If the same was used for the other matrices the spectra are not comparable since different time is optimal for them.
 Auth: As mentioned in the text, the spectra in figures 3b and 3c clearly contain the "underivatized peaks" observed in figures 3d and e. I think only the spectrum of figure 3a can be called "the lack of underivatized peaks". Please reconfirm.

comment of the reviewer
yes, the lack of peaks was too strong to write, however, the question is also relevant with „low-intensity peaks”.
Hopefully, in a future article, I will get the answer.